# Identification of microRNAs and their targets in inflorescences of an Ogura-type cytoplasmic male-sterile line and its maintainer fertile line of turnip (*Brassica rapa* ssp. *rapifera*) via high-throughput sequencing and degradome analysis

**Sue Lin**[1]*, **Shiwen Su**[2], **Libo Jin**[1], **Renyi Peng**[1], **Da Sun**[1], **Hao Ji**[1], **Youjian Yu**[3], **Jian Xu**[2]*

**1** Institute of Life Sciences, College of Life and Environmental Science, Wenzhou University, Wenzhou, China, **2** Wenzhou Vocational College of Science and Technology, Wenzhou, China, **3** College of Agriculture and Food Science, Zhejiang A & F University, Lin'an, China

* iamkari@163.com (SL); zwxuj@qq.com (JX)

**Data Availability Statement:** All of the small RNA sequencing data and degradome sequencing data

## Abstract

Cytoplasmic male sterility (CMS) is a widely used trait in angiosperms caused by perturbations in nucleus-mitochondrion interactions that suppress the production of functional pollen. MicroRNAs (miRNAs) are small non-coding RNAs that act as regulatory molecules of transcriptional or post-transcriptional gene silencing in plants. The discovery of miRNAs and their possible implications in CMS induction provides clues for the intricacies and complexity of this phenomenon. Previously, we characterized an Ogura-CMS line of turnip (*Brassica rapa* ssp. *rapifera*) that displays distinct impaired anther development with defective microspore production and premature tapetum degeneration. In the present study, high-throughput sequencing was employed for a genome-wide investigation of miRNAs. Six small RNA libraries of inflorescences collected from the Ogura-CMS line and its maintainer fertile (MF) line of turnip were constructed. A total of 120 pre-miRNAs corresponding to 89 mature miRNAs were identified, including 87 conversed miRNAs and 33 novel miRNAs. Among these miRNAs, the expression of 10 differentially expressed mature miRNAs originating from 12 pre-miRNAs was shown to have changed by more than two-fold between inflorescences of the Ogura-CMS line and inflorescences of the MF line, including 8 down- and 2 up-regulated miRNAs. The expression profiles of the differentially expressed miRNAs were confirmed by stem-loop quantitative real-time PCR. In addition, to identify the targets of the identified miRNAs, a degradome analysis was performed. A total of 22 targets of 25 miRNAs and 17 targets of 28 miRNAs were identified as being involved in the reproductive development for Ogura-CMS and MF lines of turnip, respectively. Negative correlations of expression patterns between partial miRNAs and their targets were detected. Some of these identified targets, such as squamosa promoter-binding-like transcription factor family proteins, auxin response factors and pentatricopeptide repeat-containing proteins, were previously

were submitted to the Sequence Read Archive of the NCBI under accession number PRJNA552762 (DOI: http://www.ncbi.nlm.nih.gov/sra/PRJNA552762).

**Funding:** This work was jointly supported by funds from the Special Project on Science and Technology Innovation of Seed and Seedling of Wenzhou[Z20160008] (SL), and the National Natural Science Foundation of China [31972418] (SL).

**Competing interests:** The authors have declared that no competing interests exist.

**Abbreviations:** A, adenine; AP2, APETALA 2; ARF, auxin response factor; CMS, cytoplasmic male sterility; COG, Clusters of Orthologous Group; CUC1, CUP-SHAPED COTYLEDON 1; eggNOG, orthologous groups of genes; FC, fold change; FDR, false discovery rate; FPKMs, fragments per kilobase of transcript per million mapped reads; GMUCT, genome-wide mapping of uncapped and cleaved transcripts; GRF, growth-regulating factor; KEGG, Kyoto Encyclopedia of Genes and Genomes; MF, maintainer fertile; MFEI, minimum folding energy indices; MFEs, minimum free energies; miRNAs, microRNAs; NCBI, National Center for Biotechnology Information; ncRNA, non-coding RNA; Nr, NCBI non-redundant protein; nt, nucleotide; Pfam, Homologous protein family; PPR-containing protein, pentatricopeptide repeat-containing protein; qRT-PCR, quantitative real-time PCR; rRNA, ribosomal RNA; SBP, squamosal-promoter binding protein; SD, standard deviation; snoRNA, small nucleolar RNA; snRNA, small nuclear RNA; TPM, transcripts per million; tRNA, transfer RNA; U, uracil.

reported to be involved in reproductive development in plants. Taken together, our results can help improve the understanding of miRNA-mediated regulatory pathways that might be involved in CMS occurrence in turnip.

## Introduction

Cytoplasmic male sterility (CMS) is a maternally inherited trait that is common across angiosperms and is caused by the interaction of a nuclear fertility restorer gene and a mitochondrial CMS gene, resulting in the inability to produce functional pollen [1]. CMS halts pollen development at almost all developmental stages (before, during and after meiosis of pollen mother cells) in higher plants [2]. To date, several CMS systems have been characterized, such as Ogura, Polima, Kos and NWB systems, among which there are many intra- and inter-specific variations within each cytoplasmic genotype [3–8]. Regardless of the type, the mechanism of CMS is believed to arise as a consequence of nucleus-cytoplasm incompatibility [1,2].

In most previous studies on CMS, the focus has been mainly on CMS-inducing mitochondrial genes and fertility restorer genes [9–15]. Mitochondrial dysfunction is considered to be attributed to unusual open reading frames (*orf*s) in the mitochondrial genome [10]. Many CMS-associated *orf*s have been identified and characterized in various plant species. For instance, *orf138* is associated with Ogura-CMS in Brassiceae, the co-transcription of *orf355* and *orf77* is responsible for maize CMS-S, and *orf224* co-transcribes together with *atp6* and down-regulates pollen generation in Polima-CMS of *Brassica napus* [16]. The effects of mitochondrial genes governing CMS can be suppressed by nuclear *restorer-of-fertility* (*Rf*) genes, which mostly encode pentatricopeptide repeat (PPR) proteins [12]. To date, numerous and variable *Rf*s or *Rf* candidates have been identified [17]. It is well established that the CMS phenotype is due to incompatibility resulting from the combination of mitochondrial dysfunction and a lack of *Rf* genes [1,2]. However, one fact that cannot be ignored is that nuclear genes affect the expression and function of cytoplasmic genes, as evidenced by the presence of nuclear fertility restorers, and mutant mitochondrial gene expression also contributes to retrograde regulation that fine-tunes nuclear gene expression, thus affecting pollen development [2,17–20]. Mitochondrial retrograde regulation especially manifests with variation in the stages at which pollen/anther abortion occur and the inconsistency in microsporogenesis and tapetum development processes with specific mitochondrial genes in different nuclear backgrounds [21–24]. All these findings have directed attention in recent years to studies of nucleus-cytoplasm interactions at the whole-genome level.

MicroRNAs (miRNAs) are a distinct class of single-stranded, short (~20–24 nucleotides in length), endogenously expressed non-coding RNAs (ncRNAs) that are widespread throughout the plant kingdom [25]. These small molecules are processed by Dicer-like proteins from longer RNA precursors that can form stem-loop regions [26]. Mature miRNAs are incorporated into the RNA-induced silencing complex to target their respective complementary or nearly complementary mRNAs and then act as inhibitory signals that direct mRNA cleavage or trigger translational repression [27]. MiRNAs emerge as small regulatory molecules of vital plant developmental processes, from vegetative growth to reproduction and stress responses [28–34].

Advancements in miRNA arrays and sequencing technology have led to the identification of an increasing number of miRNAs and precursor miRNAs in pollen [35–40]. In Arabidopsis, unexpectedly diverse miRNA populations belonging to 33 different miRNA families were

detected in mature pollen grains, most of which displayed an enriched expression pattern in pollen [37]. There is an increasing amount of evidence confirming that some miRNAs, such as miR159 and miR167, are master modulators of plant male sterility [41–44]. High-throughput sequencing and degradome analysis have highlighted the differential expression of miRNAs and their respective targets between CMS and fertile lines in many species [45–51]. In *B. juncea*, 47 miRNAs were differentially expressed between CMS and fertile lines, 101 in soybean, 42 in cybrid pummelo (*Citrus grandis*), and 87 in Chinese cabbage [33,45,48]. Some of these differentially expressed miRNAs were predicted to target transcription factor family proteins or functional proteins with potential roles in male gametophyte development. For instance, two novel miRNAs (novel-miR-448 and novel-miR-335) highly expressed in CMS buds of Chinese cabbage were confirmed to significantly suppress the expression of *sucrose transporter SUC1* and *H+-ATPase 6*, which perform essential roles in pollen development [48]. The discovery of miRNAs and their roles in the regulation of gene expression during pollen development shed light on the possible connection between miRNA action and CMS phenotypes. However, the regulatory network of CMS occurrence, especially the understanding of the involvement of miRNAs in CMS, is still limited.

Previously, an Ogura-CMS line 'BY10-2A' and its maintainer fertile (MF) line 'BY10-2B' of turnip (*B. rapa* ssp. *rapifera*) were characterized [52]. Mutation of mitochondrial *orf138* retroregulates the expression of nuclear genes, and interactions between them are responsible for male sterility in Ogura-CMS turnip. The first sign of disintegration shown by the anthers of Ogura-CMS line is that tapetum swells at the center of the locule during the transition from the microspore mother cells to tetrads, leading to failure of microspore development and thus complete male sterility. Using RNA sequencing analysis and bioinformatics, a large number of differentially expressed genes have been identified, which make good candidates for CMS-related genes. In this study, we further investigated the expression of miRNAs and their targets in inflorescences of the Ogura-CMS line and its MF line of turnip by high-throughput sequencing and degradome analysis. The current impact of our study is that the biogenesis of miRNAs could be regulated during retrograde signaling involved in CMS, revealing potential roles of miRNAs and their targets in regulating anther development in turnip.

## Materials and methods

### Plant materials, sample collection and total RNA isolation

An Ogura-CMS line of turnip was developed previously through consecutive back-crossing and inter-specific hybridization with *B. rapa* ssp. *chinensis* constituting the Ogura-CMS cytoplasm donor and the fertile turnip serving as the recurrent parent. The stable Ogura-CMS turnip line 'BY10-2A' and its corresponding MF line 'BY10-2B' were planted at the experimental farm of Wenzhou Vocational College of Science and Technology, Wenzhou, Zhejiang, China. All floral buds of an inflorescence from the Ogura-CMS and MF lines of turnip were collected after flowering. In each case, samples were harvested from ten individual plants and pooled, with transcriptome profiles representing '*f*' differences. Three biological replicates were performed. The samples were then immediately frozen in liquid nitrogen and stored at -70˚C until RNA isolation. Total RNA was extracted using Trizol reagent (Invitrogen, CA, USA) according to the manufacturer's protocol. A NanoPhotometer spectrophotometer (IMPLEN, CA, USA), a Qubit RNA Assay Kit in conjunction with a Qubit2.0 Fluorometer (Life Technologies, CA, USA), and an Agilent Bioanalyzer 2100 system (Agilent Technologies, CA, USA) were used to determine the RNA purity, RNA concentration, and RNA integrity, respectively, to ensure equivalent amounts of RNA samples were used for sequencing.

## Small RNA library preparation and sequencing

A total amount of 1.5 μg of RNA per sample was used as input material for small RNA library preparations using a NEBNext Ultra RNA Library Prep Kit for Illumina (NEB, CA, USA) according to the manufacturer's recommendations. Briefly, small RNA was first ligated to 3' and 5' RNA adapters. Second, the adapter-ligated RNAs were transcribed into cDNA, after which PCR was performed. Afterward, 18–30 nucleotide (nt) fragments were selected and screened via PAGE. Last, the PCR products were purified and the library was prepared. The library preparations were then used for cluster generation on a cBot Cluster Generation System of Illumina and subsequently sequenced on an Illumina HiSeq 2500 platform by Biomarker Biotechnology Corporation (Beijing, China).

## Identification of known and novel miRNAs

Clean data were obtained by removing reads containing adapters, reads containing poly-N sequences, reads of low-quality, and reads smaller than 18 nt or longer than 30 nt from the raw data. All downstream analyses were performed based on high-quality clean data as assessed by the Q20 value, Q30 value, GC-content and sequence duplication level. Ribosomal RNA (rRNA), transfer RNA (tRNA), small nuclear RNA (snRNA), small nucleolar RNA (snoRNA), other ncRNAs and repeats were then filtered and removed from the clean reads using Bowtie software [53] in conjunction with the Silva database, GtRNAdb, the Rfam database and the Repbase database. The remaining unannotated reads were then matched with assembled mRNA sequences uploaded to the Sequence Read Archive of the National Center for Biotechnology Information (NCBI) (https://www.ncbi.nlm.nih.gov/sra/PRJNA505114; accession number PRJNA505114). The matched sequences were used to detect miRNAs predicted by comparisons with previously known pre-miRNA sequences of all plants in the miRBase database (version 21.0) by miRDeep2 software [54,55]. Complete alignment of the sequences and hits with zero mismatches were considered as candidate conserved miRNAs, while the others were reserved as candidate novel miRNAs. Randfold software (version 2.1.7) and the RNAfold web server (http://rna.tbi.univie.ac.at/cgi-bin/RNAWebSuite/RNAfold.cgi) were used for secondary structure predictions of putative pre-miRNAs [54,56].

## Differential expression analysis of miRNAs

To estimate miRNA expression levels, the small RNAs were mapped back to the precursor sequences, and the read count for each miRNA was obtained from the mapping results. The frequency of miRNAs in the six libraries was normalized to the expression of transcripts per million (TPM; normalized expression = Read count × 1,000,000/Mapped reads) as proposed previously [57]. Differentially expressed miRNAs in the inflorescences of the Ogura-CMS line and its MF line were screened via the DESeq (2010) R Package. MiRNAs with a Benjamini-Hochberg false discovery rate (FDR) $\leq$ 0.05 and a $|\log_2$ fold change (FC)$| \geq 1$ were considered as differentially expressed.

## Degradome sequencing and analysis

Six degradome libraries were constructed from the inflorescences of the Ogura-CMS line and its MF line based on the genome-wide mapping of uncapped and cleaved transcripts (GMUCT) version 2.0 method as described previously [58]. In brief, poly-A-enriched mRNA was ligated with a 5'-RNA adapter, and the 5'-ligated RNA was then purified and separated from the unligated adapter by performing a second poly-A selection. Reverse transcription was performed using a primer that was a random hexamer fused to the 3'-adapter, allowing for

the adapter to be added, followed by PCR amplification. The amplified products were subsequently gel purified and subjected to deep sequencing on an Illumina HiSeq 2500 platform by Biomarker Biotechnology Corporation (Beijing, China). Clean data were obtained by trimming 5' and 3' adapters and filtering low-quality reads. The clean tags were mapped to assembled mRNA sequences (https://www.ncbi.nlm.nih.gov/sra/PRJNA505114; accession number PRJNA505114) and then annotated with rRNA, tRNA, snRNA, snoRNA, and other ncRNA from the Rfam database to obtain the unannotated tags that were used to predict subsequent degradation sites. The statistical data of the cleaved sites and target plots (t-plots) were analyzed by CleaveLand 3.0 (http://axtell-lab-psu.weebly.com/cleaveland.html) [59]. The transcript abundance was plotted for each transcript, and the heights of degradome peaks at each sliced-target transcript position were grouped into five categories according to the relative abundance of tags at the target sites [45,50]. MiRNA target sequences were searched against the NCBI non-redundant protein (Nr) database, the Swiss-Prot database and the evolutionary genealogy of genes: Non-supervised Orthologous Groups (eggNOG) database, and aligned to sequences within the Clusters of Orthologous Group (COG) database, Kyoto Encyclopedia of Genes and Genomes (KEGG) database, and Homologous protein family (Pfam) database to predict and classify their functions.

## Stem-loop and regular quantitative real-time PCR (qRT-PCR) of miRNAs and their targets

Fragments per kilobase of transcript per million mapped reads (FPKMs) expression data of miRNA targets were generated from our previous study using RNA sequencing [52]. The expression levels of miRNAs and their targets were validated by stem-loop qRT-PCR [60] and qRT-PCR, respectively. First, miRNAs were extracted from the residual plant samples for small RNA library preparations using a PureLink miRNA Isolation Kit (Thermo Fisher Scientific). For each miRNA, 1 μg of the RNA sample was reverse-transcribed with SuperScript III Reverse Transcriptase (Thermo Fisher Scientific) using a specific stem-loop reverse transcription primer (S1 Table). The reactions were then incubated for 5 min at 25˚C, followed by 50˚C for 15 min, and a final incubation at 85˚C for 5 min. The cDNA template for the miRNA targets was reverse-transcribed using SuperScript III First-Strand Synthesis SuperMix (Invitrogen, CA, USA). qRT-PCR was subsequently performed using PowerUp SYBR Green Master Mix (Applied Biosystems) on a Bio-Rad CFX384 instrument (Bio-Rad, http://www.bio-rad.com). 5S rRNA was used as an internal standard for miRNA analysis, and *ACT7* was used as an endogenous control for miRNA target analysis [61]. The melting curve was determined for the amplification specificity, and the amplified products were confirmed by sequencing. Three biological replicates and three independent technical replicates were assessed. The $2^{-\Delta\Delta Ct}$ method was applied to calculate the relative expression levels of the miRNAs and their targets [62]. All the primers used are listed in S1 and S2 Tables.

## Results

### Analysis of small RNA library data sets from inflorescences of Ogura-CMS line and its MF line of turnip

To identify miRNAs related to Ogura-CMS in turnip, six independent small RNA libraries from inflorescences collected from the Ogura-CMS line and its MF line were constructed. On average, 30,898,875 and 26,650,640 raw reads from the Ogura-CMS line and its MF line, respectively, were generated by high-throughput sequencing (S3 Table). All of the raw reads of the six libraries were uploaded to the Sequence Read Archive of the NCBI under accession

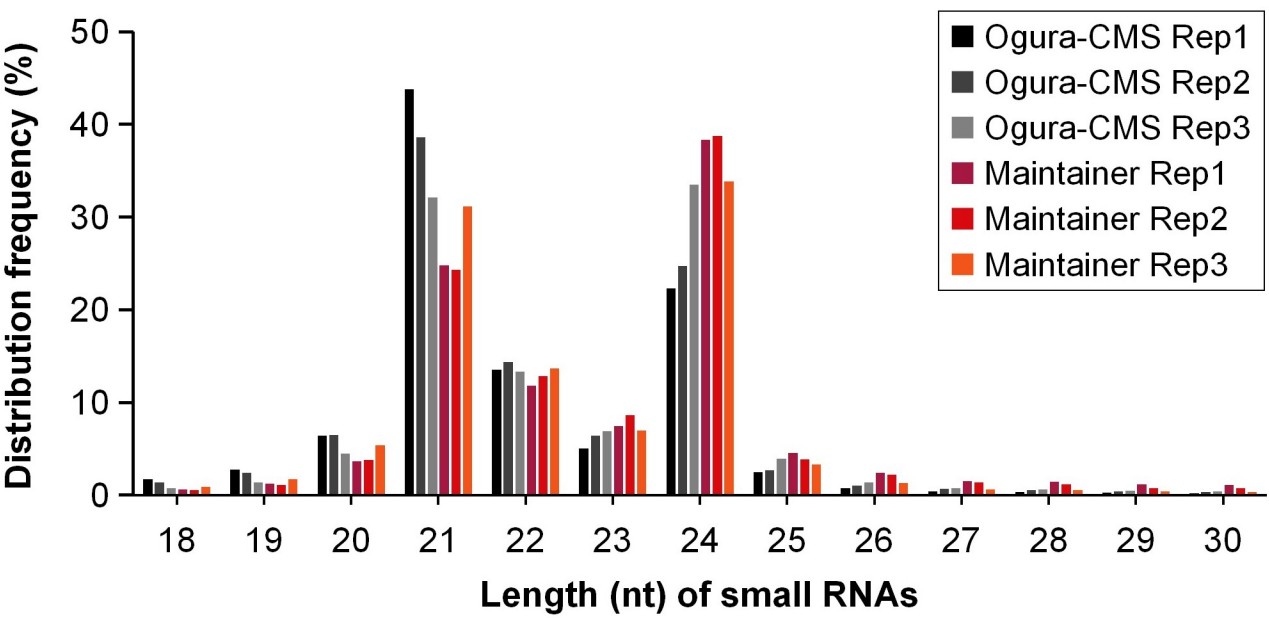

**Fig 1. Length distribution of small RNAs of inflorescences from the Ogura-CMS line and MF line of turnip.**

number PRJNA552762 (http://www.ncbi.nlm.nih.gov/sra/PRJNA552762). After removing reads containing adapters, reads containing poly-N sequences, reads of low quality, averages of 27,637,247 and 24,111,786 clean reads from the Ogura-CMS line and its MF line, respectively, were obtained, with lengths ranging from 18 nt to 30 nt, accounting for approximately 90% of the total reads (S3 Table). After filtering the rRNA, tRNA, snRNA, snoRNA, other ncRNA and repeats from the clean reads, totals of 21,524,682 and 18,774,516 unannotated reads on average, respectively, were obtained (S3 Table). Among these reads, averages of 4,471,741 and 3,673,198 small RNA sequences, respectively, were mappable, accounting for approximately 15% of the clean reads from the two lines (S3 Table). In the length distribution analysis, the majority of the small RNAs in the six libraries were 21–24 nt, with the 21 and 24 nt lengths being dominant (Fig 1), which is consistent with results of other *Brassica* species, such as Chinese cabbage (*B. rapa* ssp. *pekinensis*) and *B. juncea* [45,46,48]. The highest sequence redundancy was observed in the 21 nt long fraction of Ogura-CMS line libraries and the 24 nt long fraction of MF line libraries. In addition, the results showed that the length distribution of small RNAs was similar between the Ogura-CMS line and its MF line (Fig 1).

## Identification of known miRNAs in inflorescences of turnip

Because the whole genome of turnip is not publicly available, the mRNA transcriptome of inflorescences of turnip uploaded to the Sequence Read Archive of the NCBI (https://www.ncbi.nlm.nih.gov/sra/PRJNA505114; accession number PRJNA505114) was used as a reference sequence. To identify miRNAs involved in reproductive development in turnip, all mappable small RNA sequences aligned to the unigenes for precursor identification were compared with previously known plant miRNAs in the miRBase database (miRBase 21.0, http://www.mirbase.org/). In the present study, 120 pre-miRNAs corresponding to 89 mature 5'- or 3'-miRNAs were detected (S4 and S5 Tables).

A total of 87 pre-miRNAs corresponding to 59 mature miRNAs that have the same sequences as known pre-miRNAs in miRBase with no mismatches allowed were identified;

their read numbers among the six small RNA libraries are listed in S4 Table. The length of most known miRNAs was 21 nt, followed by 24 nt (S1A Fig). Here, there were 59 pre-miRNAs corresponding to 34 mature miRNAs originating from the Arabidopsis genome, and six mature miRNAs originating from seven pre-miRNAs came from the Brassica genome. Another 19 mature miRNAs derived from 21 pre-miRNAs identified were mapped to the genomes of other plant species. Among these 87 conserved miRNAs, ath-miR159a, ath-miR319a and ath-miR165a-3p showed very high expression levels in both the Ogura-CMS line and its MF line (S4 Table).

Among these known miRNAs, 38 miRNAs originating from 59 pre-miRNAs belonging to 30 families were detected (S6 Table). For the MIR159 family, two family members belonging to ath-miR159a were identified using deep sequencing. They shared the same mature sequences but were derived from different precursors, i.e., they originated from different loci in the turnip genome. This was also the case for several other miRNAs; a single mature miRNA sequence may originate from two to five different precursors, such as bra-miR5712, ath-miR171b-5p, ath-miR165a-3p and ath-miR394a (S4 Table). This may be attributed to the idea that some highly similar miRNA genes might be produced by a replication event from one origin sequence to another, resulting in more copies of the miRNA group. For other miR-NAs, one miRNA member was generated from a single precursor (S4 Table).

## Identification of novel miRNAs in inflorescences of turnip

To identify novel miRNAs in turnip, miRDeep2 software [54,55] and RNAfold were used to explore the secondary structures, Dicer cleavage sites and minimum free energies (MFEs) of the unannotated small RNA sequences that could be mapped to the unigenes assembled in the mRNA transcriptome sequences of inflorescences of turnip [52]. The small RNAs that mapped exactly to the assembled sequences but not to the previously known plant miRNAs in the miR-Base database were classified as candidate novel miRNAs. As a result, a total of 30 new miR-NAs derived from 33 different precursors belonging to 29 families (Tables 1 and 2) were identified as being specific to turnip in this study. Among these miRNAs, 15 new miRNA

**Table 1. Identification of new miRNA members of known miRNA families in Ogura-CMS and MF inflorescences.**

| Index | miRNA_name | miRNA_sequence | Length (nt) | MFE (kcal/mol) | MFEI | Total read count |
|---|---|---|---|---|---|---|
| 1 | N-bra-miR1222-3p | AAAGGAAUCCAUUGAAUGAGC | 21 | -70.60 | 0.7 | 863 |
| 2 | N-bra-miR1507-5p | GGCGGAGCCACAUAGAACAAGGUG | 24 | -62.50 | 0.4 | 28 |
| 3 | N-bra-miR1511-3p | AACCUGGCUCUGAUACCAUGAAGU | 24 | -59.80 | 0.3 | 117177 |
| 4 | N-bra-miR3440-3p | UUGAUUGAUCAUGGAAAGUUAGUG | 24 | -59.00 | 0.3 | 86 |
| 5 | N-bra-miR398a-5p | GGGUCGAUAUGAGAACACAUG | 21 | -60.80 | 0.3 | 3238 |
| 6 | N-bra-miR398b-5p | GGGUGACCUGAGAACACAAAACU | 23 | -49.50 | 3.8 | 68 |
| 7 | N-bra-miR398_2-5p | GGGGUGACCUGAGAACACAUG | 21 | -76.50 | 1.3 | 14 |
| 8 | N-bra-miR4415a-3p | UCGGAUUAUCAUCACAACACA | 21 | -85.90 | 2.5 | 13300 |
| 9 | N-bra-miR4415b-3p | UCGGAUUAUCAUCACAACACA | 21 | -92.50 | 3.2 | 13300 |
| 10 | N-bra-miR482-5p | AUCCAGUGAGUGGUUGUUAGAAGU | 24 | -45.90 | 0.5 | 52 |
| 11 | N-bra-miR6019-3p | UCUUCGAUCUGUAAAUGUCC | 20 | -61.40 | 3.9 | 8813 |
| 12 | N-bra-miR8005-3p | UAGGGUUUAGAAUUUAAGGUUUA | 23 | -38.05 | 0.2 | 39 |
| 13 | N-bra-miR8007-3p | UAUGCAUUUUUAGAACCUUGAGAG | 24 | -37.30 | 3.2 | 39 |
| 14 | N-bra-miR8041-3p | UUUAUAUUUCGCUAAGAACCC | 21 | -62.50 | 1.4 | 3351 |
| 15 | N-bra-miR829-5p | UUUGAAACUUUGAUCUAGAUC | 21 | -53.40 | 0.2 | 6273 |

*N* new identified.

**Table 2.  Novel miRNAs of new miRNA families identified in inflorescences of Ogura-CMS line and MF line.**

| Index | miRNA_name | miRNA_sequence | Length (nt) | MFE (kcal/mol) | MFEI | Total read count |
|---|---|---|---|---|---|---|
| 1 | N-bra-miRn1-3p | UUUUCGAUCUGUAAAUUUCCG | 21 | -88.80 | 2.9 | 379 |
| 2 | N-bra-miRn2-3p | UGUAAGAUUUAUCUCUGUAGAGGU | 24 | -17.90 | 0.1 | 21 |
| 3 | N-bra-miRn3-5p | AAAGCUUUUAACUUUGAAAAC | 21 | -69.70 | 0.8 | 6655 |
| 4 | N-bra-miRn4-3p | AGAGAUUUUUGUUACUGUUAACUA | 24 | -67.40 | 0.6 | 6899 |
| 5 | N-bra-miRn5a-3p | AAAGAUUUUUGUUACUGUUAACUG | 24 | -30.00 | 0.3 | 8145 |
| 6 | N-bra-miRn5b-3p | AAAGAUUUUUGUUACUGUUAACUG | 24 | -34.60 | 1.4 | 6261 |
| 7 | N-bra-miRn5c-5p | AAAGAUUUUUGUUACUGUUAACUG | 24 | -48.10 | 0.1 | 6263 |
| 8 | N-bra-miRn6-3p | UCAUUAAUAUCUGUUGUUCUUA | 22 | -54.60 | 0.2 | 603 |
| 9 | N-bra-miRn7-5p | AUGUAAGGAUCAAGGCUAAUCAUG | 24 | -13.00 | 0.1 | 627 |
| 10 | N-bra-miRn8-5p | UUGAGGAUCUGGGUUCAUGUC | 21 | -77.30 | 1.4 | 288 |
| 11 | N-bra-miRn9-5p | ACCAUGAGUCGAACCAGAAUG | 21 | -72.10 | 0.9 | 150 |
| 12 | N-bra-miRn10-3p | AGAGAUUUUUGUUACUGUUAACUG | 24 | -58.50 | 0.2 | 23324 |
| 13 | N-bra-miRn11-5p | UUUGGCUUGAAUCACUUCUGAAGA | 24 | -53.50 | 0.3 | 162 |
| 14 | N-bra-miRn12-5p | CGGCGAUGCGUCCUGGUCGGAUU | 23 | -111.60 | 4.9 | 45 |
| 15 | N-bra-miRn13-5p | ACUUGGAUUUUGAUGAAAUGAAUU | 24 | -54.30 | 0.6 | 53 |
| 16 | N-bra-miRn14-3p | UUGCUUAUUAGGUUCAGUGUUGGU | 24 | -27.30 | 0.4 | 76 |
| 17 | N-bra-miRn15-5p | GAGCUGUGAAGAUAAAC | 18 | -24.60 | 1.3 | 2271 |
| 18 | N-bra-miRn16-3p | AGUAAAUUAUGGAGUGGAGAUGGA | 24 | -39.50 | 2.7 | 101 |

*N* new identified.

members belonging to 13 known miRNA families were discovered in six independent small RNA libraries (Table 1). In addition, a total of 18 candidate novel miRNAs belonging to 16 novel families (Table 2) were also identified. These 33 miRNAs have not been previously reported as bra-miRNAs in miRBase. The lengths of most miRNAs were 21 nt and 24 nt (S1B Fig), which was in agreement with common characteristics of plant miRNAs. Precursors of these novel miRNAs were identified via miRDeep2 software and varied from 79 to 250 nt in length, with MFE values ranging from -13.00 to -111.60 kcal/mol, and the minimum folding energy indices (MFEIs) ranging from 0.1 to 4.9 (Tables 1 and 2; S5 Table). Most of the new mature miRNA sequences presented a uracil (U) (42.4%) or an adenine (A) (39.4%) as the first nucleotide. Like the known miRNAs, some of the novel miRNAs, such as two new miRNA members (N-bra-miR4415a-3p and N-bra-miR4415b-3p) that belonged to the MIR4415 family, shared the same mature sequence, but their precursors originated from different loci of the turnip genome. These types of miRNAs were referred to as sub-members. However, unlike the known miRNAs, the expression levels of most novel miRNAs were very low in the inflorescences of turnip (S5 Table). The secondary hairpin structures of the representative miRNA precursors to N-bra-miR8007-3p and N-bra-miRn14-3p were selected as an example that is shown in Fig 2, and others are listed in S2 Fig.

## Differential expression profiling of miRNAs in inflorescences of the Ogura-CMS line and its MF line of turnip

To ensure the reliability of the data, the count of miRNAs in the six libraries was normalized to the expression of TPM [57] and then $\log_{10}$ transformed. The correlations of the normalized data between three biological replicates for each of the two samples were then determined. As expected, the sequencing data of miRNAs in the three biological replicates showed good repeatability (all correlation coefficients ≥ 0.942) (S3 Fig), demonstrating that the results were

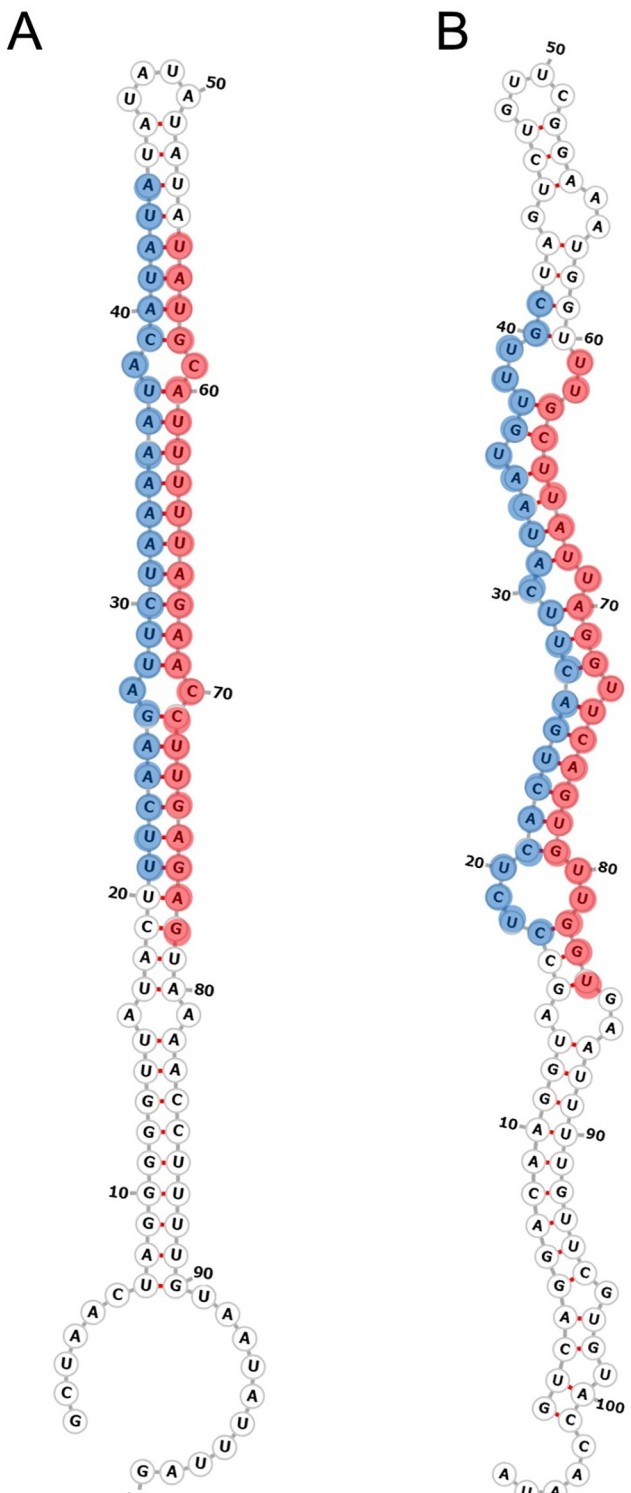

**Fig 2. Predicted secondary structure of two novel miRNAs from inflorescences of Ogura-CMS line and MF line.**
(A) N-bra-miR8007-3p. (B) N-bra-miRn14-3p. The red shaded areas indicate mature miRNAs; the blue shaded areas indicate star miRNAs.

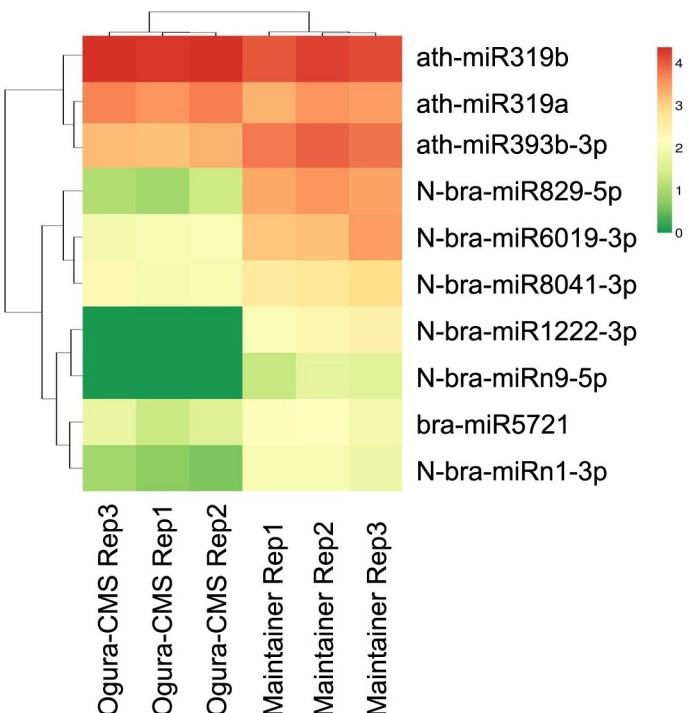

**Fig 3. Cluster heat map of differentially expressed miRNAs in Ogura-CMS and MF inflorescences of turnip.**
Differentially expressed miRNAs were filtered according to a relative fold change ≥ 2 and a Benjamini-Hochberg false
discovery rate (FDR) ≤ 0.05 after being normalized to the expression of transcripts per million (TPM). The green color
indicates a down-regulated pattern, and the red color indicates an up-regulated pattern.

reliable. The miRNAs were then subjected to differential expression analysis between the inflorescences of the Ogura-CMS line and the inflorescences of the MF line of turnip. The expression abundance of all known and novel miRNAs was compared based on filter parameters of a relative fold change ≥ 2 and a FDR ≤ 0.05 after being subjected to TPM normalization. As a result, most of the miRNAs were equally expressed; a total of 10 mature miRNAs originating from 12 pre-miRNAs showed differential expression between the inflorescences of the Ogura-CMS line and the inflorescences of the MF line according to high-throughput sequencing (Fig 3 and Table 3). Among these differentially expressed mature miRNAs, 8 miRNAs belonging to 8 miRNA families were significantly down-regulated in the Ogura-CMS line compared with the MF line, whereas 2 miRNAs belonging to the MIR319 family were up-regulated, according to the normalized sequence reads (Table 3). Notably, among the 8 down-regulated miRNAs, N-bra-miR1222-3p and N-bra-miRn9-5p were expressed specifically in the MF line.

Stem-loop qRT-PCR was conducted to validate the expression profiles of differentially expressed mature miRNAs. The results for most of the miRNAs were in agreement with those of the sequencing data (Fig 4; S4 and S5 Tables). However, the expression levels for ath-miR319a were inconsistent between the stem-loop qRT-PCR and small RNA sequencing technology. The expression of ath-miR319a was found to not significant differ between the inflorescences of the Ogura-CMS line and the inflorescences of the MF line according to stem-loop qRT-PCR, but was upregulated in the Ogura-CMS line with a $|\log_2 FC| = 1.285$ based on high-throughput sequencing analysis. This inconsistent trend possibly occurred because of the difference in sensitivity between the stem-loop qRT-PCR and high-throughput sequencing

**Table 3. List of differential expressed miRNAs between Ogura-CMS and MF inflorescences of turnip.**

| Index | miRNA_name | ID[a] | FDR | log$_2$ fold change (Ogura-CMS line/MF line) | regulated |
|---|---|---|---|---|---|
| 1 | ath-miR319a | TRINITY_DN13545_c0_g1_616 | 0.000 | 1.285 | up |
| | | TRINITY_DN13545_c0_g1_617 | | | |
| 2 | ath-miR319b | TRINITY_DN18748_c0_g1_2521 | 0.000 | 1.175 | up |
| 3 | ath-miR393b-3p | TRINITY_DN22944_c2_g5_7998 | 0.000 | -1.589 | down |
| | | TRINITY_DN22944_c2_g5_7999 | | | |
| 4 | bra-miR5721 | TRINITY_DN27641_c0_g1_38070 | 0.012 | -1.158 | down |
| 5 | N-bra-miR6019-3p | TRINITY_DN12113_c0_g1_395 | 0.000 | -3.824 | down |
| 6 | N-bra-miRn1-3p | TRINITY_DN12113_c0_g1_398 | 0.000 | -3.752 | down |
| 7 | N-bra-miR829-5p | TRINITY_DN17538_c0_g1_1994 | 0.000 | -7.252 | down |
| 8 | N-bra-miR1222-3p | TRINITY_DN22766_c4_g1_6954 | 0.000 | * | down |
| 9 | N-bra-miRn9-5p | TRINITY_DN27155_c1_g1_34381 | 0.000 | * | down |
| 10 | N-bra-miR8041-3p | TRINITY_DN46409_c0_g1_46129 | 0.002 | -1.588 | down |

[a] assemble sequence ID containing the miRNA precursor sequence.

* specifically expressed miRNAs in the MF inflorescences of turnip.

technology. Additionally, the expression patterns for each biological replicate showed high reproducibility, indicating the high reliability of the results of our deep sequencing.

## Identification of miRNA target genes in inflorescences of the Ogura-CMS line and its MF line of turnip using degradome analysis

A degradome sequencing analysis was performed to validate the miRNA targets as miRNAs function by regulating their target genes and especially by degrading their target mRNAs in

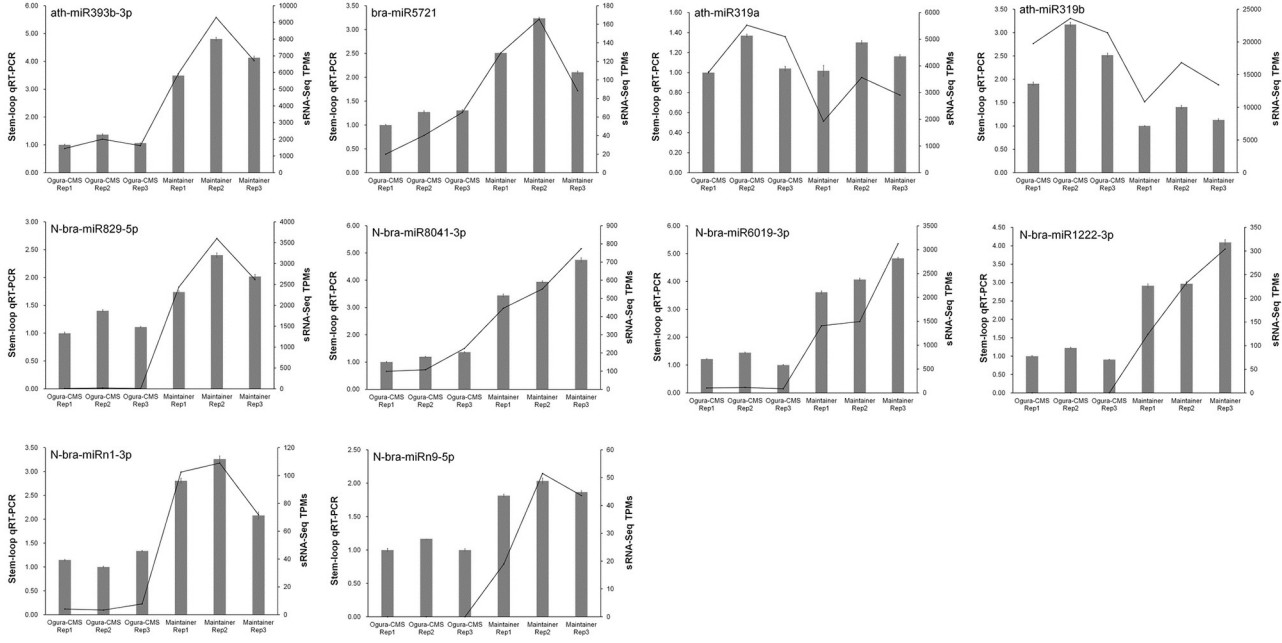

**Fig 4. Relative expression detection of miRNAs in Ogura-CMS inflorescences and MF inflorescences using stem-loop qRT-PCR.** The columns indicate the relative expression levels of miRNAs. 5S rRNA was used as an endogenous control. The results were generated from three biological replicates and three independent technical replicates, and the error bars indicate the mean ± SD (standard deviation). The lines show the normalized transcripts per million (TPM) expression data of miRNAs generated by high-throughput sequencing.

**Table 4. Analysis of degradome sequences from inflorescences of Ogura-CMS line and MF line in turnip.**

| Category | Ogura-CMS | | | Maintainer | | |
|---|---|---|---|---|---|---|
| | Rep1 | Rep2 | Rep3 | Rep1 | Rep2 | Rep3 |
| Clean reads | 12548651 | 12367523 | 11253643 | 12596323 | 12525611 | 10395432 |
| Unique reads | 4697547 (100%) | 4094011 (100%) | 4541644 (100%) | 4610371 (100%) | 4113495 (100%) | 4049399 (100%) |
| Mapped | 2767694 (58.92%) | 2291813 (55.98%) | 2742845 (60.39%) | 2675598 (58.03%) | 1999041 (48.6%) | 2191984 (54.13%) |
| rRNA | 4681 | 6058 | 4214 | 4496 | 6914 | 5458 |
| tRNA | 8 | 16 | 7 | 3 | 16 | 15 |
| snoRNA | 177 | 286 | 172 | 155 | 204 | 188 |
| snRNA | 0 | 0 | 0 | 0 | 0 | 0 |
| Other | 330 | 470 | 293 | 308 | 416 | 366 |

plants [48]. Six independent degradome libraries derived from inflorescences of Ogura-CMS line and its MF line were constructed. After removing the adapter sequences and low-quality reads from the raw reads, averages of 12,056,606 and 11,839,122 clean reads from the Ogura-CMS line and its MF line, respectively, were obtained (Table 4). These clean reads included 4,444,400 and 4,257,755 unique reads, respectively, of which 2,600,784 (58.4%) and 2,288,874 (53.6%), respectively, were perfectly matched to the unigenes assembled in the mRNA transcriptome sequencing of the inflorescences of turnip. After BLAST searches of the Rfam database and filtering out a small proportion of the hits that were annotated as ncRNAs, such as rRNA, tRNA, snoRNA, and snRNA, the remaining reads were further analyzed to identify miRNA targets.

A total of 22 targets for 25 miRNAs (23 known miRNAs and 2 novel miRNAs) and 17 targets for 28 miRNAs (26 known miRNAs and 2 novel miRNAs) were identified as being involved in the reproductive development of the Ogura-CMS and MF lines of turnip, respectively (Table 5). Based on the distribution of cleavage sites near raw sequence tags, the cleavage sites of identified miRNA targets could be divided into two categories: category '0' and category '1' (Table 5), among which category '0' is described as > 1 raw read at the position with abundance at the position equal to the maximum on the transcript and only one maximum on the transcript, and category '1' is defined as > 1 raw read at the position with abundance at the position equal to the maximum on the transcript and more than one maximum position on the transcript [45,50]. Six representative miRNAs and their corresponding targets were selected, and t-plots were constructed, in which the red line indicates the cleavage site of each transcript (Fig 5). Among these targets, most were identified for known miRNAs, and only two were for two novel miRNAs (N-bra-miR4415a-3p and N-bra-miR4415b-3p) that accumulated to a relatively high expression level (Table 5 and S5 Table). However, the target genes of most identified miRNAs could not be detected in the present degradome analysis. Unfortunately, for the vast majority of the novel miRNAs, their targets could not be identified.

According to the annotation analysis of miRNA targets, the miRNAs targeted different genes involved in a wide variety of biological functions, including several transcription factors that play important roles in gene regulation, such as homeobox-leucine zipper proteins, squamosal-promoter binding proteins (SBPs), auxin response factors (ARFs), APETALA 2 (AP2) and growth-regulating factors (GRFs) (Table 5). Some of these transcription factor targets, such as *AP2*, *TOE3*, *CUP-SHAPED COTYLEDON 1* (*CUC1*), may be related to CMS or anther development. Some non-transcription factor targets were further identified as miRNA targets in our study, such as PPR-containing proteins and disease resistance proteins. Additionally, some target genes have unknown functions (Table 5).

**Table 5. Target genes identified by degradome sequencing in inflorescences of Ogura-CMS line and MF line.**

| miRNA_name | Target gene | Target description | Alignment score | Cleavage Site | Binding sites | Category | Ogura-CMS line | Maintainer line |
|---|---|---|---|---|---|---|---|---|
| ath-miR165a-3p | TRINITY_DN27480_c0_g3 | Homeobox-leucine zipper protein REVOLUTA; transcription factor | 3 | 1146 | 1135–1155 | 0 | Y[a)] | Y |
| | TRINITY_DN26258_c1_g1 | Homeobox-leucine zipper protein ATHB-9; transcription factor | 3 | 947 | 936–956 | 0 | Y | Y |
| | TRINITY_DN22934_c0_g1 | Homeobox-leucine zipper protein ATHB-9; transcription factor | 3 | 1514 | 1503–1523 | 0 | Y | Y |
| ath-miR172a | TRINITY_DN26572_c1_g1 | Floral homeotic protein APETALA 2 (AP2); transcription factor | 2 | 3169 | 3158–3178 | 0 | Y | Y |
| | TRINITY_DN22377_c3_g5 | Function unkown | 2 | 200 | 189–209 | 1 | N[b)] | Y |
| | TRINITY_DN26572_c0_g1 | AP2-like ethylene-responsive transcription factor TOE3; transcription factor | 2.5 | 1550 | 1538–1559 | 0 | Y | Y |
| ath-miR156a-5p | TRINITY_DN23417_c1_g4 | Squamosa promoter-binding-like protein 2; transcription factor | 1 | 2236 | 2226–2245 | 0 | Y | Y |
| | TRINITY_DN23417_c1_g13 | Squamosa promoter-binding-like protein 2; transcription factor | 1 | 1040 | 1030–1049 | 1 | Y | Y |
| | TRINITY_DN23946_c2_g2 | Squamosa promoter-binding-like protein 6; transcription factor | 1 | 881 | 871–890 | 1 | Y | N |
| ath-miR172a | TRINITY_DN22377_c3_g5 | Function unkown | 2 | 200 | 189–209 | 1 | N | Y |
| | TRINITY_DN26572_c0_g1 | AP2-like ethylene-responsive transcription factor TOE3; transcription factor | 3 | 1550 | 1538–1559 | 0 | Y | Y |
| | TRINITY_DN26572_c1_g1 | Floral homeotic protein APETALA 2; transcription factor | 2.5 | 3169 | 3158–3178 | 0 | Y | Y |
| ath-miR172a | TRINITY_DN26572_c1_g1 | Floral homeotic protein APETALA 2; transcription factor | 2.5 | 3169 | 3158–3178 | 0 | Y | Y |
| | TRINITY_DN26572_c0_g1 | AP2-like ethylene-responsive transcription factor TOE3; transcription factor | 3 | 1550 | 1538–1559 | 0 | Y | Y |
| | TRINITY_DN22377_c3_g5 | Function unkown | 2 | 200 | 189–209 | 1 | N | Y |
| ath-miR165a-3p | TRINITY_DN27480_c0_g3 | Homeobox-leucine zipper protein REVOLUTA; transcription factor | 3 | 1146 | 1135–1155 | 0 | Y | Y |
| | TRINITY_DN22934_c0_g1 | Homeobox-leucine zipper protein ATHB-9; transcription factor | 3 | 1514 | 1503–1523 | 0 | Y | Y |
| | TRINITY_DN26258_c1_g1 | Homeobox-leucine zipper protein ATHB-9; transcription factor | 3 | 947 | 936–956 | 0 | Y | Y |
| ath-miR165a-3p | TRINITY_DN27480_c0_g3 | Homeobox-leucine zipper protein REVOLUTA; transcription factor | 3 | 1146 | 1135–1155 | 0 | Y | Y |
| | TRINITY_DN22934_c0_g1 | Homeobox-leucine zipper protein ATHB-9; transcription factor | 3 | 1514 | 1503–1523 | 0 | Y | Y |
| | TRINITY_DN26258_c1_g1 | Homeobox-leucine zipper protein ATHB-9; transcription factor | 3 | 947 | 936–956 | 0 | Y | Y |
| ath-miR165a-3p | TRINITY_DN27480_c0_g3 | Homeobox-leucine zipper protein REVOLUTA; transcription factor | 3 | 1146 | 1135–1155 | 0 | Y | Y |
| | TRINITY_DN26258_c1_g1 | Homeobox-leucine zipper protein ATHB-9; transcription factor | 3 | 947 | 936–956 | 0 | Y | Y |
| | TRINITY_DN22934_c0_g1 | Homeobox-leucine zipper protein ATHB-9; transcription factor | 3 | 1514 | 1503–1523 | 0 | Y | Y |

(*Continued*)

**Table 5.** (Continued)

| miRNA_name | Target gene | Target description | Alignment score | Cleavage Site | Binding sites | Category | Ogura-CMS line | Maintainer line |
|---|---|---|---|---|---|---|---|---|
| ath-miR156a-5p | TRINITY_DN23417_c1_g4 | Squamosa promoter-binding-like protein 2; transcription factor | 1 | 2236 | 2226–2245 | 0 | Y | Y |
| | TRINITY_DN23946_c2_g2 | Squamosa promoter-binding-like protein 6; transcription factor | 1 | 881 | 871–890 | 1 | Y | N |
| | TRINITY_DN23417_c1_g13 | Squamosa promoter-binding-like protein 2; transcription factor | 1 | 1040 | 1030–1049 | 1 | Y | Y |
| ath-miR156a-5p | TRINITY_DN23946_c2_g2 | Squamosa promoter-binding-like protein 6; transcription factor | 1 | 881 | 871–890 | 1 | Y | N |
| | TRINITY_DN23417_c1_g13 | Squamosa promoter-binding-like protein 2; transcription factor | 1 | 1040 | 1030–1049 | 1 | Y | Y |
| | TRINITY_DN23417_c1_g4 | Squamosa promoter-binding-like protein 2; transcription factor | 1 | 2236 | 2226–2245 | 0 | Y | Y |
| ath-miR858a | TRINITY_DN22845_c1_g6 | MYB domain protein 111; transcription factor | 3 | 363 | 352–372 | 0 | Y | Y |
| ath-miR157a-5p | TRINITY_DN23417_c1_g4 | Squamosa promoter-binding-like protein 2; transcription factor | 2 | 2237 | 2226–2246 | 0 | Y | Y |
| | TRINITY_DN23417_c1_g13 | Squamosa promoter-binding-like protein 2; transcription factor | 2 | 1041 | 1030–1050 | 1 | N | Y |
| ath-miR157a-5p | TRINITY_DN23417_c1_g13 | Squamosa promoter-binding-like protein 2; transcription factor | 2 | 1041 | 1030–1050 | 1 | N | Y |
| | TRINITY_DN23417_c1_g4 | Squamosa promoter-binding-like protein 2; transcription factor | 2 | 2237 | 2226–2246 | 0 | Y | Y |
| zma-miR2275a-5p | TRINITY_DN27219_c0_g3 | Disease resistance protein RPS6 | 2 | 864 | 853–873 | 1 | Y | N |
| | TRINITY_DN24805_c2_g4 | Disease resistance protein RPS6 | 3 | 580 | 569–589 | 1 | Y | N |
| | TRINITY_DN25134_c1_g6 | Putative disease resistance protein | 0 | 754 | 743–763 | 0 | Y | N |
| | TRINITY_DN23795_c2_g1 | Disease resistance protein RPS6 | 2 | 74 | 63–83 | 0 | Y | N |
| | TRINITY_DN27657_c1_g1 | Putative disease resistance protein | 2 | 40 | 29–49 | 0 | Y | Y |
| zma-miR2275a-5p | TRINITY_DN27219_c0_g3 | Disease resistance protein RPS6 | 2 | 864 | 853–873 | 1 | Y | N |
| | TRINITY_DN23795_c2_g1 | Disease resistance protein RPS6 | 2 | 74 | 63–83 | 0 | Y | N |
| | TRINITY_DN27657_c1_g1 | Putative disease resistance protein | 2 | 40 | 29–49 | 0 | Y | Y |
| | TRINITY_DN24805_c2_g4 | Disease resistance protein RPS6 | 3 | 580 | 569–589 | 1 | Y | N |
| | TRINITY_DN25134_c1_g6 | Putative disease resistance protein | 0 | 754 | 743–763 | 0 | Y | N |
| ath-miR5654-5p | TRINITY_DN25443_c0_g1 | Pentatricopeptide repeat (PPR)-containing protein | 3 | 207 | 196–216 | 1 | Y | Y |
| ath-miR5654-5p | TRINITY_DN25443_c0_g1 | Pentatricopeptide repeat (PPR)-containing protein | 3 | 207 | 196–216 | 1 | Y | Y |
| ath-miR160a-5p | TRINITY_DN24234_c1_g5 | Auxin response factor 10; transcription factor | 2 | 1667 | 1656–1676 | 0 | Y | Y |
| | TRINITY_DN24778_c1_g7 | Function unkown | 0 | 191 | 180–200 | 1 | N | Y |
| ath-miR172a | TRINITY_DN22377_c3_g5 | Function unkown | 2 | 200 | 189–209 | 1 | N | Y |
| | TRINITY_DN26572_c0_g1 | AP2-like ethylene-responsive transcription factor TOE3; transcription factor | 2.5 | 1550 | 1538–1559 | 0 | Y | Y |
| | TRINITY_DN26572_c1_g1 | Floral homeotic protein APETALA 2; transcription factor | 2 | 3169 | 3158–3178 | 0 | Y | Y |
| gma-miR1507c-5p | TRINITY_DN27484_c2_g1 | Pentatricopeptide repeat (PPR)-containing protein | 2 | 603 | 589–612 | 1 | Y | N |
| ath-miR160a-5p | TRINITY_DN24234_c1_g5 | Auxin response factor 10; transcription factor | 2 | 1667 | 1656–1676 | 0 | Y | Y |
| | TRINITY_DN24778_c1_g7 | Function unkown | 0 | 191 | 180–200 | 1 | N | Y |

(*Continued*)

**Table 5.** (Continued)

| miRNA_name | Target gene | Target description | Alignment score | Cleavage Site | Binding sites | Category | Ogura-CMS line | Maintainer line |
|---|---|---|---|---|---|---|---|---|
| ath-miR160a-5p | TRINITY_DN24778_c1_g7 | Function unkown | 0 | 191 | 180–200 | 1 | N | Y |
| | TRINITY_DN24234_c1_g5 | Auxin response factor 10; transcription factor | 2 | 1667 | 1656–1676 | 0 | Y | Y |
| gma-miR1518 | TRINITY_DN27162_c0_g1 | Protein VARIATION IN COMPOUND TRIGGERED ROOT growth response | 0 | 111 | 100–120 | 0 | Y | Y |
| | TRINITY_DN25949_c1_g3 | Nicotinate phosphoribosyltransferase | 1.5 | 237 | 226–246 | 0 | Y | N |
| ath-miR396a-5p | TRINITY_DN24651_c0_g1 | Growth-regulating factor 9; transcription factor | 3 | 590 | 579–600 | 0 | Y | Y |
| ath-miR172a | TRINITY_DN22377_c3_g5 | Function unkown | 3 | 200 | 189–209 | 1 | N | Y |
| | TRINITY_DN26572_c1_g1 | Floral homeotic protein APETALA 2; transcription factor | 1 | 3169 | 3158–3178 | 0 | Y | Y |
| | TRINITY_DN26572_c0_g1 | AP2-like ethylene-responsive transcription factor TOE3; transcription factor | 3 | 1550 | 1539–1559 | 0 | Y | Y |
| ath-miR157a-5p | TRINITY_DN23417_c1_g4 | Squamosa promoter-binding-like protein 2; transcription factor | 2 | 2237 | 2226–2246 | 0 | Y | Y |
| | TRINITY_DN23417_c1_g13 | Squamosa promoter-binding-like protein 2; transcription factor | 2 | 1041 | 1030–1050 | 1 | N | Y |
| ath-miR164a | TRINITY_DN25728_c2_g4 | Protein CUP-SHAPED COTYLEDON 1 | 3 | 359 | 348–368 | 0 | Y | Y |
| | TRINITY_DN25728_c2_g6 | Protein CUP-SHAPED COTYLEDON 1 | 3 | 381 | 370–390 | 0 | Y | Y |
| N-bra-miR4415a-3p | TRINITY_DN27162_c0_g1 | Protein VARIATION IN COMPOUND TRIGGERED ROOT growth response | 1 | 111 | 100–120 | 0 | Y | Y |
| | TRINITY_DN25949_c1_g3 | Nicotinate phosphoribosyltransferase | 1 | 237 | 226–246 | 0 | Y | N |
| N-bra-miR4415b-3p | TRINITY_DN27162_c0_g1 | Protein VARIATION IN COMPOUND TRIGGERED ROOT growth response | 1 | 111 | 100–120 | 0 | Y | Y |
| | TRINITY_DN25949_c1_g3 | Nicotinate phosphoribosyltransferase | 1 | 237 | 226–246 | 0 | Y | N |

[a] Y indicates that target gene could be detected;

[b] N indicates that target gene could not be detected.

## Expression of miRNA targets in inflorescences of the Ogura-CMS line and its MF line of turnip

To test the causal relationship between the level of miRNAs and their target expression, four selective miRNAs and their corresponding cleavage targets identified in the degradome sequencing analysis were satisfactorily verified by stem-loop qRT-PCR and regular qRT-PCR, respectively. As expected, the expression levels of the targets of ath-miR156a-5p (TRINITY_DN18236_c0_g1_2308), two sub-members of ath-miR157a-5p (TRINITY_DN23691_c0_g1_12492 and TRINITY_DN40654_c0_g1_45037) and ath-miR5654-5p (TRINITY_DN24251_c3_g2_15972) were higher in the CMS line than in the MF line, which contrasted with their corresponding miRNA data (Fig 6). ath-miR156a-5p targets SBPs (TRINITY_DN23417_c1_g4, TRINITY_DN23417_c1_g13 and TRINITY_DN23946_c2_g2), two sub-members of ath-miR157a-5p also target SBPs (TRINITY_DN23417_c1_g4 and TRINITY_DN23417_c1_g13), and ath-miR5654-5p targets TRINITY_DN25443_c0_g1 (PPR-

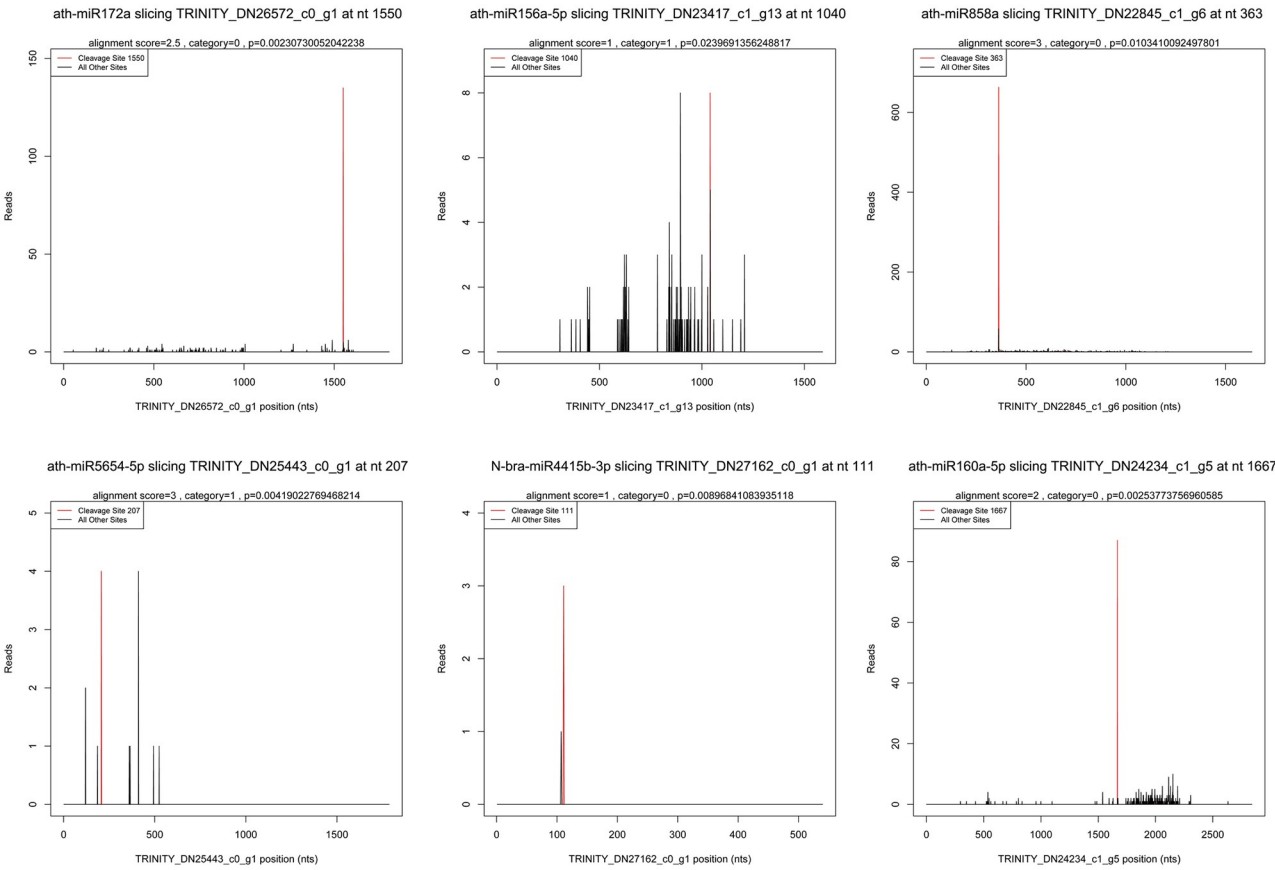

**Fig 5. T-plots of representative identified miRNA targets in inflorescences of Ogura-CMS line and MF line.** The X axis indicates the site position of miRNA target sequences, and the Y axis indicates the abundance of raw tags. The red peak indicates the predicted cleavage site and $p \leq 0.05$.

containing protein) (Table 5). Taken together, the results showed that there are negative correlations between the expression patterns of miRNAs and their targets.

## Discussion

The predominant role of miRNAs in plant biological processes and responses to environmental stresses has directed attention in recent years to plant miRNA research [31,32]. With the development of in silico tools and innovative techniques, such as small RNA deep sequencing coupled with degradome analysis and miRNA arrays, a substantial number of miRNAs have been identified in Arabidopsis, rice and many other plant species [63,64]. However, the focus of the research community is still concentrated on model and key plant species [34]. Although the vast majority of miRNA families are conserved in angiosperms, miRNA members and their expression levels vary by plant species [27]. MiRNA identification should be expanded to non-model and lesser-explored plant species. Turnip is one of the most important local root vegetables in China and has been consumed for thousands of years [65]. In the present study, using high-throughput sequencing, a total of 120 pre-miRNAs corresponding to 89 mature miRNAs were identified in inflorescences of the Ogura-CMS line and its MF line of turnip (S4 and S5 Tables). Among these identified miRNAs, 33 miRNAs were novel (Tables 1 and 2). Based on the miRNA family analysis, 38 known miRNAs originating from 59 pre-miRNAs were subjected to 30 families (S6 Table). Some of the identified miRNAs are highly conserved

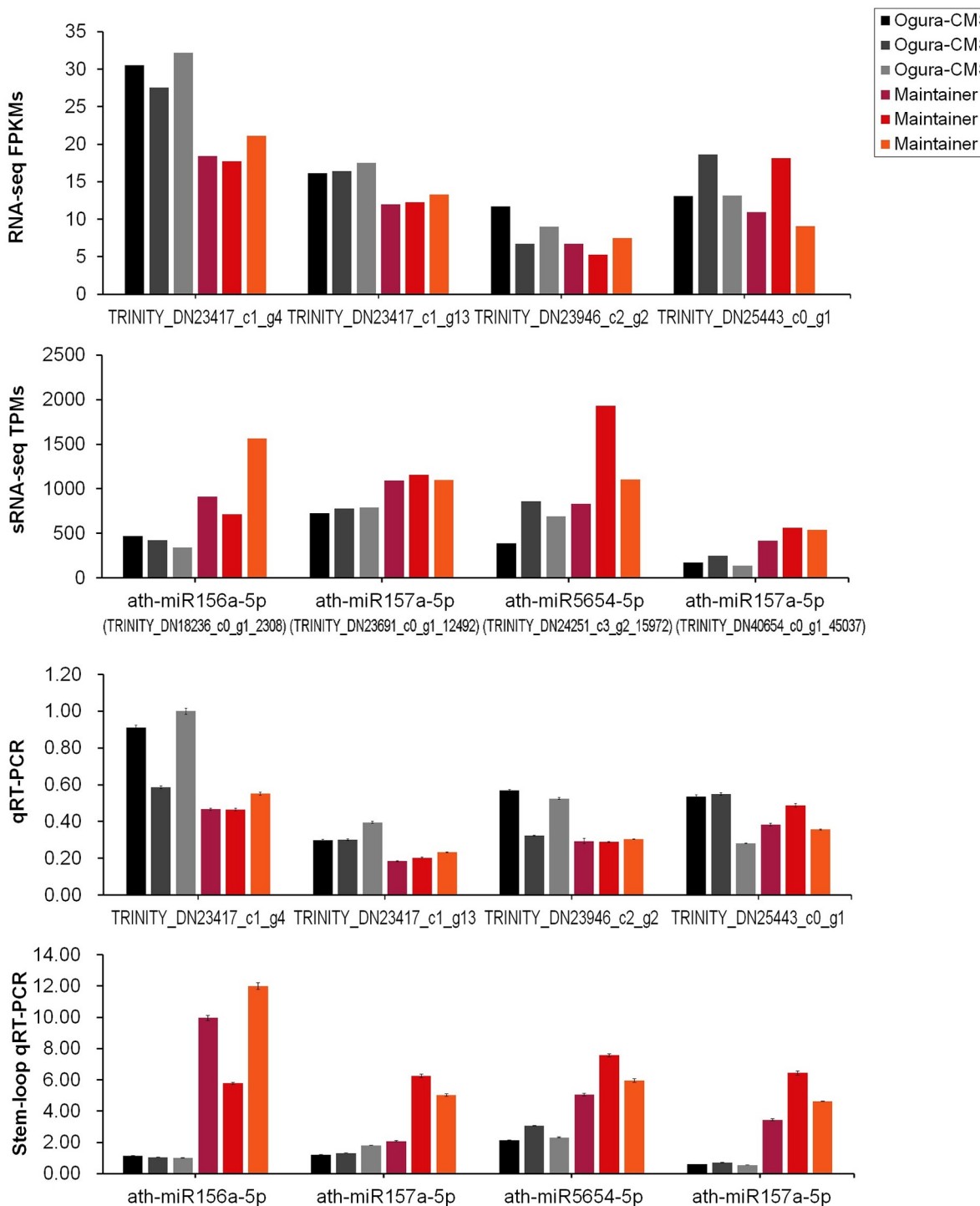

**Fig 6. Expression of miRNAs and their corresponding targets in inflorescences of Ogura-CMS line and MF line.** The normalized transcripts per million (TPM) expression data of miRNAs were generated from small RNA sequencing. The fragments per kilobase of transcript per million mapped reads (FPKMs) expression data of miRNA targets were generated from RNA-sequencing [52]. 5S rRNA was used as an internal standard for miRNA analysis, and *ACT7* was used as an endogenous control for miRNA target analysis. The results were generated from three biological replicates and three independent technical replicates, and the error bars indicate the mean ± SD (standard deviation).

among diverse angiosperms, such as miR156, miR160, miR164, miR172 and miR396, while some are species specific, such as miR5718, which is specific to *B. campestris* [27,66].

The participation of miRNAs in pollen development is becoming increasingly evident [66]. In the case of CMS, the regulatory network responsible for pollen generation refers to the rearrangements of mitochondrial CMS-inducing genes and their interactions with nuclear genes [10]. Although the exact nature of miRNAs with respect to CMS in plants remains elusive, discovery of miRNAs with possible implications for CMS induction is intriguing in this respect. The recent explosion of high-throughput sequencing has enabled genome-wide discovery of these regulatory molecules and their targets, implying their potential involvement in CMS [67]. Using this method combined with bioinformatic analysis, researchers have successively identified differentially expressed miRNAs and their targets between CMS lines and fertile lines of rice, *B. juncea*, cybrid pummelo (*Citrus grandis*) and many other species [45–51]. In the present study, 10 differentially expressed mature miRNAs originating from 12 pre-miRNAs in turnip were identified with more than two-fold changes in expression between inflorescences of the Ogura-CMS line and inflorescences of the MF line (Fig 3 and Table 3). Although direct evidence substantiating miRNAs as causative agents for CMS phenotypes remains unavailable, the differences in miRNA biogenesis in the Ogura-CMS line in our study may partly reflect mitochondrial retrograde regulation of the nuclear transcriptome.

Using high-throughput sequencing coupled with degradome analysis, we identified a total of 22 targets for 25 miRNAs and 17 targets for 28 miRNAs involved in reproductive development for the Ogura-CMS line and its MF line of turnip, respectively (Table 5). Plant miRNAs have been implicated in the negative regulation of the stability of their target genes by RNA cleavage and translational repression, of which directing target cleavage is the best-known and predominant mode of miRNA action [27]. The most common targets for CMS-related miRNAs are genes encoding transcription factors. MiRNA-mediated regulation of transcription factors affects various downstream related genes, suggesting a core role of the involvement of miRNAs in gene regulatory networks [68]. In the present study, several transcription factor targets were identified, such as homeobox-leucine zipper proteins, SBPs, ARFs, AP2s, GRFs and MYB domain proteins (Table 5). Some of these transcription factor targets resemble those of previous studies with respect to their defined or inferred roles in regulating reproductive development processes. For instance, miR156 and miR157 are well known for controlling anther development by targeting SBP-like genes [69,70]. miR396 targets *GRF* genes, including *GRF9* [68]. In addition, miR172-mediated regulation of *AP2* and *AP2*-like genes such as *TOE3* is essential for floral organ identity in Arabidopsis [71–73]. Perturbing miR164-directed repression of *CUC1* disrupts floral development [74]. Although conclusive evidence ascertaining the linkage between *ARF10* and reproductive development remains obscure, overexpression of two other *ARF* genes (*ARF16* and *ARF17*) free from miR160 regulation was reported to result in aberrant flowers with floral organ defects and reduced fertility [75–77].

Apart from transcription factors, some functional protein targets were also identified for the miRNAs in our study, indicating a role for miRNAs in regulating protein stability. PPR-containing proteins constitute an important class of fertility restorer proteins that are required for generating a functional male gametophyte in CMS plants [14,19]. PPR-containing proteins have been previously confirmed as targets of miR158, miR161, miR475 and miR476 [69,78,79]. Here, miR5654 and miR1507 were also predicted to target PPR-containing proteins (Table 5). In addition, other protein targets were also identified, such as disease resistance proteins and phosphoribosyltransferase (Table 5). Together, all of these transcription factor and functional protein target prediction analysis results indicate that some of the miRNAs are involved in anther development and male sterility, leading to a possible connection between miRNA action and Ogura-CMS phenotype.

In the present study, the discovery of miRNAs and their targets helps to improve the understanding of the contribution of miRNAs in CMS and the influence of mitochondrial CMS-inducing genes on global transcriptome changes, although, the effects of miRNAs on CMS occurrence and the exact roles of miRNA-mediated regulation of transcription factors or other downstream related genes in CMS induction is still demanding complementing genetic technologies to substantiate the findings of our present study.

## Conclusion

In this study, high-throughput sequencing and degradome analysis were employed to enable genome-wide discovery of conserved and novel miRNAs and their targets possibly involved in anther development in turnip. Comprehensive analysis of miRNAs and their targets in inflorescences of the Ogura-CMS line and its MF line of turnip suggested that mutation of mitochondrial *orf138* leads to the fine-tuned expression of miRNAs, which may further participate in the regulatory network of CMS occurrence by regulating their target genes.

## Supporting information

**S1 Fig. Length distribution of known and novel miRNAs in inflorescences from Ogura-CMS line and MF line.** (A) Known miRNAs. (B) Novel miRNAs.
(TIF)

**S2 Fig. Predicted secondary structures of novel miRNAs from inflorescences of Ogura-CMS line and MF line.** The red shaded areas indicate mature miRNAs; the blue shaded areas indicate star miRNAs.
(TIF)

**S3 Fig. Correlation coefficiencies between three biological replicates for the sequencing data of miRNAs.**
(TIF)

**S1 Table. Specific primers used for stem-loop quantitative real-time PCR (qRT-PCR) validation of miRNAs in turnip.**
(XLSX)

**S2 Table. Primers designed for quantitative real-time PCR (qRT-PCR) validation of miRNA targets in turnip.**
(XLSX)

**S3 Table. Analysis of small RNA sequences from inflorescences of Ogura-CMS line and MF line of turnip.**
(XLSX)

**S4 Table. Known miRNAs identified in inflorescences of Ogura-CMS line and MF line of turnip.**
(XLSX)

**S5 Table. Novel miRNAs identified in inflorescences of Ogura-CMS line and MF line of turnip.**
(XLSX)

**S6 Table. Identification of known miRNA members of known miRNA families in Ogura-CMS and MF inflorescences.**
(XLSX)

## Acknowledgments

We thank Dr. Heng Dong (Hangzhou Normal University, China) for the critical comments and editing on this manuscript.

## Author Contributions

**Data curation:** Sue Lin, Shiwen Su.

**Funding acquisition:** Sue Lin.

**Investigation:** Da Sun, Hao Ji, Youjian Yu.

**Project administration:** Sue Lin, Jian Xu.

**Resources:** Sue Lin, Jian Xu.

**Validation:** Libo Jin, Renyi Peng.

**Writing – original draft:** Sue Lin.

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
