## [Decision Letter · Decision Letter 0]

18 May 2020

PONE-D-20-10719

Identification of microRNAs and their targets in inflorescences of Ogura-type cytoplasmic male-sterile line and its maintainer fertile line in turnip (*Brassica rapa* ssp. *rapifera*) by high-throughput sequencing and degradome analysis

PLOS ONE

Dear Dr. Lin,

Thank you for submitting your manuscript to PLOS ONE. After careful consideration, we feel that it has merit but does not fully meet PLOS ONE’s publication criteria as it currently stands. Therefore, we invite you to submit a revised version of the manuscript that addresses the points raised during the review process.

i) The language needs serious improvements; ii) The inconsistence in the results should be addressed; iii) The sequencing data should be deposited into public databases, such as NCBI GEO/SRA and include the accession IDs in the manuscript.

We would appreciate receiving your revised manuscript by Jul 02 2020 11:59PM. To enhance the reproducibility of your results, we recommend that if applicable you deposit your laboratory protocols in protocols.io, where a protocol can be assigned its own identifier (DOI) such that it can be cited independently in the future. For instructions see: http://journals.plos.org/plosone/s/submission-guidelines#loc-laboratory-protocols

We look forward to receiving your revised manuscript.

Kind regards,

Yun Zheng, Ph.D

Academic Editor

PLOS ONE
---

## [Author Response · Author response to Decision Letter 0]

5 Jun 2020

Responses to the academic editor:

1. The language needs serious improvements.

Response: We have had our paper professionally edited for the English language by a manuscript proofreading service named Springer Nature Author Services (https://secure.authorservices.springernature.com). The readability of our paper is greatly improved and we hope English is accessible to readers this time.

2. The inconsistence in the results should be addressed.

Response: We have taken the concerns seriously and have tried to do our best to address the inconsistence in the results. Please find the revised manuscript resubmitted to PLOS ONE and see our revisions and responses below.

3. The sequencing data should be deposited into public databases, such as NCBI GEO/SRA and include the accession IDs in the manuscript.

Response: All of the small RNA sequencing data and degradome-sequencing data have been submitted to the Sequence Read Archive of NCBI, and the accession number PRJNA552762 (DOI: http://www.ncbi.nlm.nih.gov/sra/PRJNA552762) has been included in our revised manuscript.

4. Response: No, we would not want to make changes to our financial disclosure.

This work was jointly supported by funds from the Special Project on Science and Technology Innovation of Seed and Seedling of Wenzhou [Z20160008] (SL), and the National Natural Science Foundation of China [31972418] (SL).

5. To enhance the reproducibility of your results, we recommend that if applicable you deposit your laboratory protocols in protocols.io, where a protocol can be assigned its own identifier (DOI) such that it can be cited independently in the future. For instructions see: http://journals.plos.org/plosone/s/submission-guidelines#loc-laboratory-protocols.

Response: The protocols for total RNA isolation, small RNA library preparations, stem-loop qRT-PCR and qRT-PCR used in our manuscript are regular, and the methods were performed following manufacturer’s recommendations. Here, we have no unique laboratory protocol that needs to be deposited in protocols.io in our manuscript.

A rebuttal letter that responds to each point raised by the academic editor and reviewer(s). This letter should be uploaded as separate file and labeled 'Response to Reviewers'.

A marked-up copy of your manuscript that highlights changes made to the original version. This file should be uploaded as separate file and labeled 'Revised Manuscript with Track Changes'.

An unmarked version of your revised paper without tracked changes. This file should be uploaded as separate file and labeled 'Manuscript'.

Response: A rebuttal letter labeled 'Response to Reviewers', a marked-up copy labeled 'Revised Manuscript with Track Changes', and an unmarked version labeled 'Manuscript' have been included when submitting our revised manuscript.

Please ensure that your manuscript meets PLOS ONE's style requirements, including those for file naming. The PLOS ONE style templates can be found at https://journals.plos.org/plosone/s/file?id=wjVg/PLOSOne_formatting_sample_main_body.pdf and https://journals.plos.org/plosone/s/file?id=ba62/PLOSOne_formatting_sample_title_authors_affiliations.pdf

Response: We ensure that our manuscript meets PLOS ONE's style requirements, including those for file naming.

Response: Before resubmission, we have uploaded our figure files to the PACE digital diagnostic tool. PACE generated figure files that meet PLOS requirements have been downloaded and used in our revised manuscript. Please find the adjusted figures in the revised manuscript resubmitted to PLOS ONE.

----------

Responses to reviewer #1:

1. The manuscript has identified some conserved miRNAs and novel miRNAs in turnip inflorescences collected from Ogura-CMS line and its maintainer fertile. Author has finally identified 12 differentially expressed miRNAs between Ogura-CMS inflorescences and its MF inflorescence, and most importantly, degradome had been used and verified some targerts for these miRNAs, however, there were some minor English language errors, such as below:

1.1. line 45: “coorperation” usually used as between people’s action, here, auther decribed the two genes’ interaction, I advise to change as “interaction” could much be better.

Response: “cooperation” on line 45 in previous submission has been replaced by “interaction” on line 47 in our revised manuscript.

In addition, we have had our paper professionally edited for the English language by a manuscript proofreading service named Springer Nature Author Services (https://secure.authorservices.springernature.com). We hope English is accessible this time.

1.2. Line 53:” “retrograde” means degenerate, backward or get worse, so I advise to replace as “reverse”.

Response: Thank you for your advice. Mitochondrial perturbations can cause changes in the expression of nuclear genes. This type of communication between mitochondrial and the nucleus is called mitochondrial retrograde regulation (Liao and Butow, 1993, Cell, 72, 61-71). We have listed several CMS-related articles in which the word “retrograde” is used.

[1] Kubo T, Arakawa T, Honma Y, Kitazaki K. What does the molecular genetics of different types of restorer-of-fertility genes imply? Plants. 2020;9: 361.

[2] Singh S, Dey SS, Bhatia R, Kumar R, Behera TK. Current understanding of male sterility systems in vegetable Brassicas and their exploitation in hybrid breeding. Plant Reprod. 2019;32: 231-256.

[3] Chen Z, Zhao N, Li S, Grover CE, Nie H, Wendel JF, et al. Plant mitochondrial genome evolution and cytoplasmic male sterility. Crit Rev Plant Sci. 2017;36: 55–69.

[4] Horn R, Gupta KJ, Colombo N. Mitochondrion role in molecular basis of cytoplasmic male sterility. Mitochondrion. 2014;19: 198–205.

[5] Roads, DM. Plant mitochondrial retrograde regulation. In: Kempken, F, editor. Plant Mitochondria. New York: Springer; 2011. pp. 411–437.

[6] Yang J, Zhang M, Yu J. Mitochondrial retrograde regulation tuning fork in nuclear genes expressions of higher plants. J. Genet. Genomics. 2008;35: 65-71.

[7] Lee B, Lee H, Xiong L, Zhu JK. A mitochondrial complex I defect impairs cold-regulated nuclear gene expression. Plant Cell. 2002;14: 1235–1251.

So we also use “retrograde”. In addition, some of the references mentioned above were included when we used “retrograde” in our manuscript.

1.3. Line 69 “dagradome” should be “degradome”.

Response: “dagradome” on line 69 in previous submission has been replaced by “degradome” on line 81 in our revised manuscript.

2. Introduction: Please tell more about the research progress in miRNA and CMS in plants.

Response: More information about the research progress in miRNA and CMS in plants has been incorporated into the Introduction section on lines 54-62 of page 3, lines 77-79 of page 4, and 82-88 of pages 4-5.

3. Results: Table 1 described the data qaulities got from the machines, which may not be the main or key results of this manucript, I suggest the author to put this table into the supplementary files.

Response: Thank you for your suggestion. We have put Table 1 into the supplementary files in our revised manuscript.

4. In conclusion, I think this manuscript had done a well job for characterization for genome-wide analysis of miRNAs between MF and normal turnip inflorescences. I think it could be accepted by this journal after English revised by the native language specialits.

Response: We really appreciate your positive comments on our manuscript. We have taken your comments seriously and have had our paper professionally edited for the English language by a manuscript proofreading service named Springer Nature Author Services (https://secure.authorservices.springernature.com). We hope that the revisions are satisfactory.

----------

Responses to reviewer #2:

1. Generally, this study was well designed and analyzed, the manuscript is well written and understandable.

Response: We really appreciate your positive comments on our manuscript.

2. It is a little confusing about the differentially expressed miRNAs. 12 miRNAs were identified as differentially expressed miRNAs, however ath-miR319a and ath-miR393b-3p were counted twice (Figure 3, table 4), is it because each of them has two precursors? sRNA-Seq can only detected mature miRNA expression, I suggest changing number to 10, in such case, you can still check the expression of different miRNA precursor.

Response: It is a critical point. Thanks for your comments. In our study, to identify miRNAs, the matched sequences were compared with previously known pre-miRNA sequences of all plants from miRBase database. These 12 differentially expressed miRNAs are corresponding to 10 mature miRNAs, among which, TRINITY_DN13545_c0_g1_616 and TRINITY_DN13545_c0_g1_617 share the same mature miRNA (ath-miR319a) sequences but are derived from different precursors. This is also the case for ath-miR393b-3p, a single mature miRNA sequence originates from two different precursors (TRINITY_DN22944_c2_g5_7998 and TRINITY_DN22944_c2_g5_7999). In our revised manuscript, we rephrased the result as “10 mature miRNAs originating from 12 pre-miRNAs showed differential expression…” on lines 274-275 of page 17 or “10 differentially expressed mature miRNAs originating from 12 pre-miRNAs …” on line 33 of page 2 and line 388 of page 36. In Fig 3 and Table 3 (previously named Table 4), the number of differentially expressed mature miRNAs was changed to 10.

3. The disagreement of miR319a expression between qRT-PCR and sequencing probably be due to authors just checked 1 precursor (TRINITY_DN13545_c0_g1_616), how about the other precursor?

Response: The stem-loop qRT-PCR was conducted to validate the expression profiles of differentially expressed mature miRNAs in our study. This method is designed to detect and quantify mature miRNAs in a fast, specific, accurate and reliable manner (Varkonyi-Gasic and Hellens, 2011. Quantitative stem-loop RT-PCR for detection of microRNAs. In: Kodama H, Komamine A, editors. RNAi and Plant Gene Function Analysis. Totowa, NJ: Humana Press. pp. 145–157). So we think that the disagreement of miR319a expression between qRT-PCR and sequencing is not due to the check of 1 precursor. However, in previous submission, “(TRINITY_DN13545_c0_g1_616)” on line 260 of page 18 and in Fig 4, and “TRINITY_DN22944_c2_g5_7998” in Fig 4 may be a little confusing. In our revised manuscript, we deleted “(TRINITY_DN13545_c0_g1_616)” and “TRINITY_DN22944_c2_g5_7998” on the text and in Fig 4. In addition, we also deleted the assembly sequence ID containing the miRNA precursor sequence in S1 Table. Because these probably make the understanding of the results complicated.

4. Where do the data of target gene expression “RNA-Seq FPKMs” come from in Figure 6? I did not see any information in the main text.

Response: Previously, we investigated the morphological characteristics of the Ogura-CMS line ‘BY10-2A’ and its maintainer fertile (MF) line ‘BY10-2B’ of turnip, and conducted a detailed inflorescence transcriptome analysis for the Ogura-CMS line and MF line using RNA sequencing technology (Lin et al., 2019, PLoS ONE, 14(6), e0218029). The RNA sequencing data was uploaded to the Sequence Read Archive of the NCBI (DOI: https://www.ncbi.nlm.nih.gov/sra/PRJNA505114; accession number PRJNA505114). The data of target gene expression “RNA-Seq FPKMs” in Figure 6 came from the article that we previously published in PLoS ONE (Lin et al., 2019, PLoS ONE, 14(6), e0218029). In our revised manuscript, the information about the data of target gene expression “RNA-Seq FPKMs” was incorporated into the Materials and Methods section on lines 173-174 of page 9 as “Fragments per kilobase of transcript per million mapped reads (FPKMs) expression data of miRNA targets were generated from our previous study using RNA sequencing [52].” In addition, the legend for Fig 6 was revised as “Fig 6. Expression of miRNAs and their corresponding targets in inflorescences of Ogura-CMS line and MF line. The normalized transcripts per million (TPM) expression data of miRNAs were generated from small RNA sequencing. The fragments per kilobase of transcript per million mapped reads (FPKMs) expression data of miRNA targets were generated from RNA-sequencing [52]. 5S rRNA was used as an internal standard for miRNA analysis, and ACT7 was used as an endogenous control for miRNA target analysis. The results were generated from three biological replicates and three independent technical replicates, and the error bars indicate the mean ± SD (standard deviation).”, on lines 357-363 of page 34.

5. Authors should give a little information about Ogura-CMS line ‘BY10-2A’ and mitochondrial ORF138 in introduction or material part, so that readers can understand their suggestion that “mutation of mitochondrial ORF138 leads to fine-tuned expression of miRNAs”.

Response: The information about Ogura-CMS line ‘BY10-2A’ and mitochondrial orf138 was incorporated into the Introduction section on lines 93-94 of page 5 as “Mutation of mitochondrial orf138 retro-regulates the expression of nuclear genes, and interactions between them are responsible for male sterility in Ogura-CMS turnip”.

6. Figures resolution is low, can not see the gene and miRNA ID.

Response: In our revised manuscript, figures were prepared with a resolution of 300 dpi. In addition, we have uploaded our figure files to the PACE digital diagnostic tool before resubmission. PACE generated figure files that meet PLOS requirements have been downloaded and used in our revision. Please find the adjusted figures in the revised manuscript resubmitted to PLOS ONE.

7. The grammar of this sentence (Page 18) need be corrected “However, expression of ath-miR319a (TRINITY_DN13545_c0_g1_616) was found to have no significant differences between Ogura-CMS inflorescences and its MF inflorescences using stem-loop qRT-PCR, however, was upregulated in Ogura-CMS with |log2 FC| = 1.285 based on the high-throughput sequencing analysis”

Response: The sentence on page 18 in previous submission, “However, expression of ath-miR319a (TRINITY_DN13545_c0_g1_616) was found to have no significant differences between Ogura-CMS inflorescences and its MF inflorescences using stem-loop qRT-PCR, however, was upregulated in Ogura-CMS with |log2 FC| = 1.285 based on the high-throughput sequencing analysis”, has been replaced by “However, the expression levels for ath-miR319a were inconsistent between the stem-loop qRT-PCR and small RNA sequencing technology. The expression of ath-miR319a was found to not significant differ between the inflorescences of the Ogura-CMS line and the inflorescences of the MF line according to stem-loop qRT-PCR, but was upregulated in the Ogura-CMS line with a |log2 FC| = 1.285 based on high-throughput sequencing analysis.”

8. Sentence on page 37, “Besides PPR-containing proteins, other protein targets including disease resistance proteins, phosphoribosyltransferase, etc (Table 6)” need be fixed.

Response: Sentence on page 37 in previous submission, “Besides PPR-containing proteins, other protein targets including disease resistance proteins, phosphoribosyltransferase, etc (Table 6)” has been replaced by “In addition, other protein targets were also identified, such as disease resistance proteins and phosphoribosyltransferase (Table 5).”

9. Page 43, line 393, instead of “.” , “,” should be used before “although”

Response: “.” on line 393 of page 43 in previous submission has been replaced by “,” on line 421 of page 37 in our revised manuscript.

10. It is better change “RNA-Seq TPMs” to “sRNA-Seq TPMs” in Figure 6.

Response: “RNA-Seq TPMs” has been replaced by “sRNA-Seq TPMs” in Figure 6 in our revised manuscript. In addition, “RNA-Seq TPMs” has also been replaced by “sRNA-Seq TPMs” in Figure 4.

11. Table 4 (and other supplemental tables), miRNA_ID is not accurate, it actually represent the assembly sequence ID containing the miRNA precursor sequence.

Response: “miRNA_ID” in Table 4 and other supplemental tables (S4, S5 and S6 Tables) in previous submission has been replaced by “IDa)” in our revised manuscript. In addition, notes about “a)” have been added under the tables as “a) assembly sequence ID containing the miRNA precursor sequence”.

12. The sequencing data should be deposited for public availability.

Response: All of the small RNA sequencing data and degradome sequencing data have been submitted to the Sequence Read Archive of the NCBI, and the accession number PRJNA552762 (DOI: http://www.ncbi.nlm.nih.gov/sra/PRJNA552762) has been included in our revised manuscript.

---

## [Decision Letter · Decision Letter 1]

15 Jul 2020

Identification of microRNAs and their targets in inflorescences of an Ogura-type cytoplasmic male-sterile line and its maintainer fertile line of turnip (Brassica rapa ssp. rapifera) via high-throughput sequencing and degradome analysis

PONE-D-20-10719R1

Dear Dr. Lin,

We’re pleased to inform you that your manuscript has been judged scientifically suitable for publication and will be formally accepted for publication once it meets all outstanding technical requirements.

Kind regards,

Yun Zheng, Ph.D

Academic Editor

PLOS ONE

Additional Editor Comments (optional):

Reviewers' comments:

Reviewer's Responses to Questions

**Comments to the Author**

1. If the authors have adequately addressed your comments raised in a previous round of review and you feel that this manuscript is now acceptable for publication, you may indicate that here to bypass the “Comments to the Author” section, enter your conflict of interest statement in the “Confidential to Editor” section, and submit your "Accept" recommendation.

Reviewer #1: All comments have been addressed

Reviewer #2: All comments have been addressed

2. Is the manuscript technically sound, and do the data support the conclusions?

Reviewer #1: Yes

Reviewer #2: Yes

3. Has the statistical analysis been performed appropriately and rigorously? 

Reviewer #1: Yes

Reviewer #2: Yes

4. Have the authors made all data underlying the findings in their manuscript fully available?

Reviewer #1: Yes

Reviewer #2: Yes

5. Is the manuscript presented in an intelligible fashion and written in standard English?

Reviewer #1: Yes

Reviewer #2: Yes

6. Review Comments to the Author

Reviewer #1: The author has anwered and improved all the questions in the revised manuscript, I advise to accept for publication.

Reviewer #2: The authors have addressed all the questions, this version has been greatly improved.

in the "marked-up copy", there are several places need be corrected:

Page 3, line 63, the 1st word “and” should be deleted.

Page 8, line 163 and other places, “adapter” should be “adaptor”. The sentence “5’ -ligated RNA was then purified away and separated from the unligated adapter by performing a second poly-A selection’ should be “5’ -ligated RNA was then purified again by performing a second-round poly-A selection”.

7. PLOS authors have the option to publish the peer review history of their article (what does this mean?). If published, this will include your full peer review and any attached files.

Reviewer #1: **Yes: **Jun Yang

Reviewer #2: No

---

## [Editor Report · Acceptance letter]

20 Jul 2020

PONE-D-20-10719R1 

Identification of microRNAs and their targets in inflorescences of an Ogura-type cytoplasmic male-sterile line and its maintainer fertile line of turnip (Brassica rapa ssp. rapifera) via high-throughput sequencing and degradome analysis 

Dear Dr. Lin:

I'm pleased to inform you that your manuscript has been deemed suitable for publication in PLOS ONE. Congratulations! Your manuscript is now with our production department. 

Kind regards, 

on behalf of

Dr. Yun Zheng 

Academic Editor

PLOS ONE